# Core Deficits and Eating Behaviors in Children with Autism: The Role of Executive Function

**DOI:** 10.3390/nu17243854

**Published:** 2025-12-10

**Authors:** Yufei Liu, Kelong Cai, Qiyi Wang, Shuai Guo, Shuqiao Meng, Kai Qi, Yifan Shi, Zhiyuan Sun, Xuan Xiong, Aiguo Chen

**Affiliations:** 1Department of Physical Education, Gdansk University of Physical Education and Sport, 80-336 Gdansk, Poland; yufei.liu@awf.gda.pl (Y.L.);; 2Nanjing Sport Institute, Nanjing 210014, China; 3College of Physical Education, Yangzhou University, Yangzhou 225127, China; 4Faculty of Sport and Leisure, Guangdong Ocean University, Zhanjiang 524088, China; 5Department of Physical Education, Xidian University, Xi’an 710126, China; 6Department of Physical Education, Nanjing University, Nanjing 210008, China

**Keywords:** autism, social impairments, restrictive and repetitive behaviors, executive function, eating behaviors

## Abstract

**Background**: Children with autism spectrum disorder (ASD) frequently experience eating-related behavioral difficulties; however, the relationships among these difficulties, core ASD deficits, and executive function remain poorly understood. The present study examined how core ASD characteristics—restrictive and repetitive behaviors (RRBs) and social impairments—relate to eating behaviors, including food approach and avoidance tendencies. In addition, the study explored whether executive function serves as a mediating mechanism underlying these associations. **Methods**: A total of 184 children aged 3–12 years participated in this cross-sectional study. All variables were measured using parent-reported questionnaires, and data were analyzed through path modeling. The Social Responsiveness Scale–Second Edition (SRS-2) and the Repetitive Behavior Scale–Revised (RBS-R) were employed to assess social impairments and RRBs, respectively. Eating behaviors—comprising food approach and food avoidance dimensions—were evaluated with the Children’s Eating Behavior Questionnaire (CEBQ), while executive function was measured using the Child Executive Functioning Inventory (CHEXI). **Results**: The analysis revealed significant associations between RRBs and both food approach and food avoidance behaviors in children with ASD. Crucially, follow-up regression analyses specifying RRBs subtypes showed that Stereotyped Behavior independently predicted both food approach (β = 0.305, *p* < 0.001) and avoidance (β = 0.217, *p* = 0.002), while Compulsive Behavior specifically predicted food avoidance (β = 0.173, *p* = 0.021). Furthermore, executive function appeared to serve as a potential mediator in these relationships, suggesting that impairments in executive control may partially explain how repetitive behaviors influence eating patterns. Although social impairments showed weaker direct associations with eating behaviors, they indirectly affected both food approach and avoidance behaviors through deficits in executive function, highlighting the complex interplay among behavioral, cognitive, and social domains in ASD. **Conclusions**: These findings indicate that RRBs—one of the core characteristics of ASD—can predict children’s eating behaviors and are concurrently linked to two seemingly opposite eating patterns. Both social impairments and RRBs appear to influence eating behaviors through executive dysfunction. This study provides new insights into the mechanisms underlying atypical eating behaviors in children with ASD and identifies executive function as a promising target for interventions aimed at improving eating-related outcomes in this population.

## 1. Introduction

Autism Spectrum Disorder (ASD) is a complex neurodevelopmental condition characterized by persistent deficits in social communication and restrictive, repetitive behaviors (RRBs) [1]. In addition to its core Deficits, ASD is frequently associated with disturbances in eating behavior and other behavioral difficulties [2]. These issues encompass a broad spectrum of eating-related difficulties, including atypical eating patterns, abnormal food preferences and motivations, and dysregulated eating—such as emotional eating [3], feeding difficulties [4], obesity [5], and picky eating [6,7]. Moreover, evidence indicates that these eating behavior problems occur at significantly higher rates in individuals with ASD than in typically developing (TD) children. These atypical eating behaviors not only contribute to obesity and growth retardation [8] but may also adversely affect cognitive development. Considering the high prevalence, persistence, and potentially severe consequences of eating behavior problems in children with ASD, systematic investigation in this domain holds critical clinical and theoretical value.

### 1.1. Core Symptoms and Eating Behaviors in Children with Autism

The higher prevalence of eating behavior problems in children with ASD compared with TD children suggests that ASD core deficits may play a crucial role in shaping eating behaviors [2]. Previous studies have shown that these core symptoms can predict disordered eating patterns [9], contributing to both binge eating and food avoidance behaviors. Moreover, longitudinal evidence highlights that core deficits are significant long-term predictors of binge eating behaviors [10]. It is also noteworthy that children are generally more prone to eating behavior difficulties than adults [11,12]. ASD core symptoms have been consistently linked to a range of maladaptive eating behaviors in children, such as food selectivity, emotional eating, reduced food intake, and increased mealtime disruptive behaviors [13,14]. Taken together, these findings underscore a strong association between ASD core deficits and eating behaviors in children, emphasizing the need for further exploration of the underlying mechanisms to inform effective behavioral interventions.

In previous studies examining the predictive role of core autism deficits on eating behaviors, the two primary domains of autism spectrum disorder were often merged into a single construct. While instruments such as the Broad Autism Phenotype Questionnaire (BAPQ), the Social Responsiveness Scale–Second Edition (SRS-2), and the Social and Communication Disorders Checklist (SCDC) can be used to assess overall autistic traits [11,14,15], combining these distinct domains—or relying on one to represent the whole construct—oversimplifies their underlying mechanisms. Because the two domains reflect fundamentally different behavioral and neurocognitive processes, merging them may obscure domain-specific effects and introduce interpretive bias. Therefore, clearly delineating how each domain uniquely relates to eating behaviors is crucial for developing precise and targeted intervention strategies for children with ASD.

Most existing studies on eating behaviors have primarily focused on obesity-related behaviors such as binge eating [3], whereas research on food avoidance behaviors has been relatively scarce. Nonetheless, growing evidence indicates that these two types of behaviors are closely interrelated and may even transform into one another under certain conditions [16]. As a result, increasing attention has been directed toward food avoidance behaviors, and recent studies have underscored the equal importance of examining restrictive eating patterns [17]. According to the Children’s Eating Behavior Questionnaire [18], eating behaviors can be divided into eight dimensions: food responsiveness, enjoyment of food, emotional overeating, desire to drink, satiety responsiveness, slowness in eating, emotional undereating, and food fussiness. In the present study, the first four dimensions were classified as food approach behaviors, and the latter four as food avoidance behaviors. Both categories were analyzed as dependent variables to emphasize their relative importance. This classification framework has been widely adopted and validated in previous international studies, confirming its scientific reliability and conceptual soundness [19,20]. Based on this framework, the present study hypothesizes that the core deficits of ASD—social impairments and RRBs—are significantly associated with eating behaviors in children with ASD.

### 1.2. The Relationship Between Executive Function and Eating Behavior in the Core Deficits of Autism

The determinants of eating behavior are complex, involving interactions among core deficits, cognitive functions, sensory sensitivities, and other factors [21,22]. Emerging evidence suggests that executive function impairments—which are themselves shaped by core deficits —may influence eating behaviors and act as a mediating mechanism [23,24]. However, the current evidence base remains fragmented, and no empirical study has yet examined a complete mediating pathway from core deficits, through executive function, to specific eating behaviors.

The executive function impairment theory of ASD provides a theoretical foundation for the proposed mediating pathway. This theory posits that abnormalities in the prefrontal cortex and related neural circuits lead to deficits in key cognitive processes—such as inhibitory control and cognitive flexibility—that underpin a wide range of behaviors [25], extending to various aspects of daily life, including eating behavior. Building upon this theoretical foundation, Demetriou proposed a cognitive-behavioral framework of executive function in ASD, emphasizing that distinct components of executive function—such as inhibitory control, working memory, cognitive flexibility, and planning—play critical roles in shaping everyday behaviors [26]. However, these frameworks primarily emphasize associations between executive function and behavior and do not empirically examine executive function as a mediating mechanism linking specific core deficits to concrete behavioral outcomes. Empirical research that simultaneously considers ASD core deficits, executive function, and eating behaviors remains limited. Therefore, we hypothesize that executive function acts as a potential mediator between the core deficits of ASD (social communication difficulties and RRBs) and distinct eating behavior patterns (food approach and food avoidance).

Given the substantial challenges that children with autism spectrum disorder face in both learning and daily life, investigating the psychological mechanisms underlying their atypical eating behaviors offers a crucial foundation for developing effective interventions that promote physical health and improve overall quality of life. Building on an established theoretical framework and supported by empirical evidence, we therefore constructed and validated a path model to clarify the proposed relationships (Figure 1). In this context, the present study focuses on examining the potential links between core deficits of autism spectrum disorder and eating behaviors, with particular attention to whether executive function operates as a mediating factor in these associations.

## 2. Materials and Methods

### 2.1. Study Design

This study adopted a cross-sectional design to investigate the associations between core ASD deficits (social deficits and RRBs) and eating behaviors (food approach and food avoidance), while also examining the potential mediating role of executive function using a path analysis model. Participants were continuously recruited from the Huiying Child Development Center in Yangzhou, China. Ethical approval was obtained from the Ethics and Human Protection Committee of the Affiliated Hospital of Yangzhou University, and the study was registered in the Chinese Clinical Trial Registry (ChiCTR1900024973, ChiCTR2400087262). For the cross-sectional analyses, only baseline data from a series of studies were included; these data were collected more than 72 h before the first intervention to minimize potential confounding factors and prevent contamination from subsequent treatments. All research procedures were conducted in accordance with the latest version of the Declaration of Helsinki. Prior to participation, the study’s objectives and procedures were clearly explained to the parents of all participating children, and written informed consent was obtained from each participant’s guardian.

### 2.2. Participants

A total of 199 children with ASD were recruited for this study. The inclusion criteria were: (1) Han ethnicity; (2) diagnosis of ASD according to DSM-5 criteria; (3) age between 3 and 12 years; and (4) provision of informed consent indicating willingness to participate. The exclusion criteria were: (1) the presence of other neurological or psychiatric disorders; (2) current use of medications that affect the central nervous system; (3) physical disabilities; (4) visual or hearing impairments; and (5) prior treatment for eating disorders. Among the 199 children initially enrolled, 15 were excluded due to incomplete or incorrectly completed questionnaires, resulting in a final sample of 184 valid participants.

### 2.3. Measures

The independent variables in this study were the core symptoms of ASD, which comprised two dimensions: social deficits and RRBs. These two domains were measured using the Social Responsiveness Scale, Second Edition (SRS-2) and the Repetitive Behavior Scale–Revised (RBS-R), respectively. The dependent variable was eating behavior, divided into two subdimensions—food approach and food avoidance—and assessed with the Children’s Eating Behavior Questionnaire (CEBQ). The mediating variable was executive function, evaluated using the Child Executive Functioning Inventory (CHEXI).

Given the large number of questionnaire items (n = 167) and the moderate sample size, we used the mean scores of each subscale as manifest variables in the path model. This strategy allowed us to maintain a reasonable variable-to-sample ratio and preserve model parsimony, as modeling all items as independent indicators would have exceeded the analytical capacity of the dataset. Moreover, all scales have been thoroughly validated in Chinese children with ASD aged 3–12 and show strong factor structures and reliability, thereby supporting the use of subscale means as robust manifest indicators. Additionally, demographic and clinical variables—including sex, autism symptom severity, date of birth, and body mass index (BMI)—were collected. BMI was calculated as weight in kilograms divided by height in meters squared (kg/m^2^). Autism symptom severity was assessed using the Childhood Autism Rating Scale (CARS), which was administered by a qualified hospital-based medical professional.

#### 2.3.1. The Social Responsiveness Scale, Second Edition (SRS-2)

This study evaluated the social communication abilities of children with ASD using the parent-reported Social Responsiveness Scale–Second Edition (SRS-2). The SRS-2 is a quantitative instrument developed to assess social functioning in children with ASD aged 4–18 years, and a Chinese-adapted version of the scale was utilized in the present study [27]. The scale contains 65 items rated on a 4-point Likert scale and is divided into five subscales: Social Awareness (8 items), Social Cognition (12 items), Social Communication (22 items), Social Motivation (11 items), and Restricted Interests and Repetitive Behaviors (12 items). Each subscale score represents the mean of the items within that domain. In this scoring system, a value of 4 indicates severe behavioral difficulties, 3 indicates moderate difficulties, 2 indicates mild difficulties, and 1 indicates the absence of behavioral problems. Higher scores reflect greater symptom severity or social impairment. Caregivers completed the scale based on the child’s typical behaviors observed over the preceding three months. The Chinese version of the SRS-2 has been extensively validated and widely applied in studies involving Chinese children with ASD aged 3–12 years, consistently demonstrating excellent internal reliability, with Cronbach’s alpha coefficients above 0.8 across all subscales for both parent and teacher reports [28,29,30].

#### 2.3.2. The Repetitive Behavior Scale-Revised (RBS-R)

This study evaluated RRBs in children with ASD using the parent-reported Repetitive Behavior Scale–Revised (RBS-R), with a Chinese-adapted version applied in the current investigation [31]. The RBS-R comprises 43 items rated on a 4-point Likert scale and is divided into six subscales: Stereotyped Behavior (6 items), Self-Injurious Behavior (8 items), Compulsive Behavior (8 items), Ritualistic Behavior (6 items), Sameness Behavior (11 items), and Restricted Interests (4 items). Each subscale score represents the mean value of its corresponding items. In this scoring system, a score of 4 indicates severe behavioral problems, 3 indicates moderate problems, 2 indicates mild problems, and 1 indicates no behavioral problems. Higher scores reflect a greater severity of repetitive behaviors. Caregivers completed the scale based on the child’s typical behaviors observed over the preceding three months. The Chinese version of the RBS-R has been extensively validated and widely utilized in studies involving Chinese children with ASD aged 3–12 years. It has demonstrated satisfactory internal reliability, with Cronbach’s alpha coefficients exceeding 0.7 across all subscales [31,32,33].

#### 2.3.3. The Child Executive Functioning Inventory Scale (CHEXI)

This study evaluated executive function in children with ASD using the parent-reported Child Executive Functioning Inventory (CHEXI), with a Chinese-adapted version applied in the present research [34]. The CHEXI comprises 24 items rated on a 5-point Likert scale and includes four dimensions: Working Memory (9 items), Planning (4 items), Inhibitory Control (6 items), and Regulation (5 items). Each subscale score was calculated as the mean of its corresponding items. Responses ranged from 1 (“completely inconsistent”) to 5 (“completely consistent”), where higher scores indicated greater executive dysfunction. Caregivers completed the questionnaire based on their observations of the child’s typical behaviors during the preceding three months. The Chinese version of the CHEXI has been extensively validated and widely used in studies involving Chinese children with ASD aged 3–12 years, demonstrating robust internal consistency, with Cronbach’s alpha coefficients above 0.7 for all dimensions [35,36].

#### 2.3.4. The Children’s Eating Behavior Questionnaire (CEBQ)

This study evaluated eating behaviors in children with ASD using the parent-reported Children’s Eating Behavior Questionnaire (CEBQ), with a Chinese-adapted version applied in the present study [18]. The CEBQ consists of 35 items rated on a 5-point Likert scale and assesses two major domains of eating behavior. The first domain, food approach, includes four subscales: Food Responsiveness (5 items), Enjoyment of Food (4 items), Emotional Overeating (4 items), and Desire to Drink (3 items). The second domain, food avoidance, comprises four subscales: Satiety Responsiveness (5 items), Slowness in Eating (4 items), Emotional Undereating (4 items), and Food Fussiness (6 items). Each subscale score was calculated as the mean of its respective items. Response options ranged from 1 (“never”) to 5 (“always”), with higher scores indicating a greater frequency of the corresponding eating behavior. Caregivers completed the questionnaire based on their observations of the child’s typical eating patterns during the previous three months. The Chinese version of the CEBQ has been extensively validated and widely used in studies of Chinese children with ASD aged 3–12 years, demonstrating satisfactory internal reliability, with Cronbach’s alpha coefficients above 0.7 across all subscales [37,38,39].

### 2.4. Data Analysis

A total of 184 children with autism spectrum disorder were included in this study. Some variables contained missing data, with missing rates ranging from 0% to 7.2%. Because the missing values were dispersed and showed no systematic pattern, they were considered potentially missing at random. To address this issue, we applied multiple imputation using the “mice” package in R (version 4.4.1). This procedure helped preserve statistical power and reduce bias, thereby enhancing the robustness of the analyses. The overall analysis proceeded in four stages:

Stage 1: Descriptive statistics and one-way analyses of variance (ANOVA) were performed using SPSS version 29.0.2.0 for Mac, after confirming the normality of data distribution. This stage aimed to examine differences in core symptoms, executive function, and eating behaviors among ASD children grouped by gender, age, BMI, and symptom severity. In addition, this analysis helped identify potential confounding variables.

Stage 2: Path modeling and model fit evaluation were conducted using Amos version 26.0. Considering that the measurement instruments used in this study (SRS-2, RBS-R, CHEXI, and CEBQ) have established cross-cultural validity in ASD populations, and that their Chinese versions are widely applied among Chinese children with ASD aged 3–12 years, the mean scores of each subscale were used as manifest indicators in the model. Previously published confirmatory factor analysis (CFA) results were referenced to support construct validity.

Stage 3: For the core deficits that demonstrated significant associations with eating behavior pathways, this study utilized SPSS (version 29.0.2.0 for Mac) to construct linear regression models aimed at identifying which specific subtypes could independently predict children’s eating behaviors. Two separate regression models were established: one with the total score of “food approach” as the dependent variable and another with the total score of “food avoidance.” In both models, different subdimensions of RRBs were included as independent variables. This analytical approach allowed for the evaluation of each subdimension’s unique contribution to eating behaviors while controlling for the effects of all other RRBs subdimensions.

Stage 4: The mediating effects among variables were tested using the bootstrap method, which provides higher statistical power for detecting indirect effects compared with traditional approaches such as the Sobel test [40]. Four mediation models were developed, and the bootstrap resampling was performed 5000 times. Results were reported using 95% bias-corrected (BC) confidence intervals, and statistical significance was set at *p* < 0.05 for determining significant mediation effects.

## 3. Results

### 3.1. Characteristics of Study Participants

Table 1 presents the characteristics of all participating children with ASD, as well as the differences in core deficits, executive function, and eating behaviors across varying levels of each characteristic. The number of male participants was higher than that of females, which aligns with the known prevalence ratio of ASD (4.2:1) [41]. The mean age of participants was 6.42 years, with an average BMI of 17.13 and an average CARS score of 38.54.

Regarding the results across different characteristic levels, no significant differences were found in social ability, repetitive behaviors, executive function, food approach behaviors, or food avoidance behaviors among ASD children differing in gender, age, or BMI (Ps > 0.05). However, significant differences were observed across ASD severity levels, as defined by CARS scores (CARS < 37 indicating mild ASD, CARS ≥ 37 indicating severe ASD). Specifically, children with varying ASD severity showed significant differences in repetitive behaviors (F(1,182) = 28.98, *p* < 0.001), social ability (F(1,182) = 47.21, *p* < 0.001), executive function (F(1,182) = 219.13, *p* < 0.001), food approach behaviors (F(1,182) = 10.07, *p* = 0.002), and food avoidance behaviors (F(1,182) = 32.68, *p* < 0.001).

### 3.2. Goodness-of-Fit of the Path Model

The hypothesized relationships among the study variables were tested using a path analysis model estimated with the maximum likelihood method. Table 2 summarizes the results for eight key model fit indices. Overall, the findings indicated that the structural model achieved a satisfactory level of fit, meeting the accepted criteria for model adequacy. Specifically, the similarity indices (GFI, AGFI, IFI, CFI, and TLI) all exceeded 0.800, whereas the dissimilarity indices (SRMR and RMSEA) were below 0.080 [42,43], suggesting that the model exhibited good overall fit and supporting the robustness of the study’s results.

### 3.3. Path Analysis (Direct Effects)

Figure 2 presents the results of the path analysis. The analysis revealed that greater severity of RRBs was significantly associated with poorer executive function (γ = 0.174, B = 0.262, *p* = 0.046), as well as with higher frequencies of both food approach behaviors (γ = 0.409, B = 0.855, *p* < 0.001) and food avoidance behaviors (γ = 0.620, B = 0.750, *p* < 0.001). Additionally, lower levels of social responsiveness were significantly correlated with poorer executive function (γ = 0.467, B = 0.759, *p* < 0.001). However, the direct associations between social responsiveness and either food approach (γ = −0.140, B = −0.316, *p* = 0.175) or food avoidance behaviors (γ = −0.128, B = −0.166, *p* = 0.211) were not statistically significant. Finally, poorer executive function was significantly related to higher frequencies of both food approach behaviors (γ = 0.369, B = 0.512, *p* < 0.001) and food avoidance behaviors (γ = 0.379, B = 0.304, *p* < 0.001), suggesting a potential mediating role of executive function in the relationship between core ASD symptoms and eating behaviors.

### 3.4. Regression Analysis of RRBs Subtypes on Eating Behaviors

Given the significant direct pathways between the overall RRBs and both eating behaviors, we conducted multiple linear regression analyses to identify which specific RRBs subtypes independently predicted feeding patterns. The multicollinearity diagnostic indicated that the variance inflation factors (VIFs) ranged from 1.23 to 2.04, all well below the commonly accepted threshold of 5. These values suggest that multicollinearity was not a major concern, permitting confident interpretation of the regression coefficients and supporting the identification of each subscale’s unique association with eating behavior outcomes. All six RBS-R subscales were entered simultaneously into each model to assess their unique contributions while controlling for other subtypes. As shown in Table 3, the analysis indicated that Stereotyped Behavior [B = 0.145, t = 3.106, *p* = 0.002, 95% CI: 0.053–0.237] and Compulsive Behavior [B = 0.170, t = 2.327, *p* = 0.021, 95% CI: 0.026–0.314] were significantly associated with food avoidance behaviors. Furthermore, Stereotyped Behavior [B = 0.266, t = 4.073, *p* < 0.001, 95% CI: 0.137–0.395] was also significantly related to food approach behaviors. Although Compulsive Behavior did not show a statistically significant association with food approach, a marginal trend was observed (*p* = 0.080).

### 3.5. Mediation Analysis (Indirect Effects)

Table 4 summarizes the analysis of the indirect effects of executive function on the relationships between core ASD symptoms and eating behaviors. The results indicated that the indirect pathways from RRBs to food approach (Bias-Corrected 95% CI [0.004, 0.375]) and food avoidance behaviors (Bias-Corrected 95% CI [0.001, 0.233]) were both statistically significant. Interestingly, although the direct effects of social responsiveness on eating behaviors were not significant, the indirect pathways through executive function were significant for both food approach (Bias-Corrected 95% CI [0.138, 0.767]) and food avoidance behaviors (Bias-Corrected 95% CI [0.071, 0.470]). Collectively, these findings highlight the mediating role of executive function in linking social responsiveness and repetitive behaviors with eating behavior patterns among children with ASD.

## 4. Discussion

This study employed path analysis to systematically explore the complex relationships between core symptoms of ASD and eating behaviors, with a particular emphasis on the mediating role of executive function. The analysis identified two primary pathways. First, RRBs were directly linked to higher frequencies of both food approach and food avoidance behaviors. Specifically, Stereotyped Behavior and Compulsive Behavior were significantly associated with food avoidance, while Stereotyped Behavior was also positively related to food approach behaviors. Moreover, RRBs exerted indirect effects on these behaviors through impairments in executive function, indicating that deficits in cognitive control may partially explain the behavioral patterns observed. Second, although social deficits did not directly predict eating behaviors, their indirect influence through executive function was statistically significant. Taken together, these findings underscore the critical role of executive function as a cognitive bridge connecting core ASD symptoms to maladaptive eating behavior patterns.

### 4.1. The Influence of RBBs on Eating Behavior in Children with ASD and the Mediating Role of Executive Function

This study found that RRBs simultaneously predicted two seemingly contradictory patterns of eating behavior—food approach and food avoidance. This finding suggests that RRBs may shape both approach-oriented and avoidance-oriented eating phenotypes in children with ASD through distinct but interrelated mechanisms. Specifically, RRBs may promote food approach behaviors via sensory-seeking tendencies and reward dependence. Children with ASD often display strong fixations on specific food textures, temperatures, or flavors (e.g., accepting only crunchy foods). Such rigid sensory preferences can motivate them to actively seek foods that match their repetitive expectations [44], thereby potentially engaging reward-related neural pathways and increasing the likelihood of frequency of food approach behaviors [45]. As a potential neural correlate that warrants further investigation, this behavioral association is consistent with findings from neuroimaging research. Studies have shown that when individuals with ASD are exposed to preferred foods, they exhibit heightened activation in the striatum and prefrontal cortex that aligns with their stereotyped expectations [14,46]. Conversely, any deviation from these sensory routines may trigger significant anxiety, whereas performing repetitive eating behaviors can serve as a coping mechanism to relieve such distress [47]. Over time, this neural adaptation may establish a positive feedback loop that reinforces repetitive eating patterns, potentially leading to ritualized eating behaviors.

On the other hand, RRBs were also significantly associated with food avoidance. Path analysis revealed a strong direct effect (γ = 0.620), a finding that warrants cautious interpretation. Because both constructs were measured using parent reports, common method bias may have inflated the observed association. Nevertheless, the effect aligns with theoretical expectations. Prior studies have identified RRBs as a potential core mechanism underlying eating problems such as food selectivity and avoidant/restrictive food intake disorder (ARFID) [48,49]. Accordingly, the strong association observed in this study suggests that food avoidance may not occur independently but may instead represent a behavioral manifestation of RRBs within the routine and repetitive context of eating. This linkage is likely driven by the cognitive rigidity and sensory defensiveness characteristic of RRBs.

Specifically, similar to the mechanisms underlying food approach behaviors, children with ASD may exhibit food avoidance as a selective rejection of non-preferred foods (e.g., those with hard textures) due to repetitive behavioral tendencies. Such children often require specific and predictable conditions during mealtimes, including food preparation methods, eating schedules, or ritualized behaviors [48]. Because of their limited flexibility in adjusting expectations and behaviors, individuals with ASD may overestimate the potential risks associated with non-preferred foods (e.g., irrational avoidance of hard textures), which in turn elicits strong avoidance responses [50]. Although food approach and food avoidance behaviors appear phenotypically opposite, the present findings suggest that both may originate from the same underlying mechanism—RRBs. Thus, selective rejection of non-preferred foods and heightened sensitivity to environmental changes during eating may, in essence, serve as avoidance strategies that help individuals with ASD maintain internal sensory homeostasis and cognitive consistency. Over time, these patterns may develop into maladaptive eating behaviors such as picky eating, food refusal, or selective eating [51,52,53,54].

The mediation model confirmed that executive function serves as a key intermediary linking RRBs with both food approach and food avoidance behaviors in children with ASD. This finding aligns with the executive dysfunction theory of ASD, which proposes that RRBs are outward behavioral expressions of deficits in higher-order cognitive control, highlighting the close relationship between these constructs [24,55]. Moreover, this theoretical perspective can be extended to explain eating behavior patterns, suggesting that impairments in executive function may underlie the repetitive and rigid characteristics observed in the eating behaviors of children with ASD.

As a higher-order cognitive function, executive function may engage multiple interconnected processes when mediating the relationship between RRBs and eating behaviors in children with ASD. Individuals exhibiting pronounced RRBs often display deficits in inhibitory control, particularly when confronted with highly rewarding or preferred foods. Such impairments make it difficult to suppress strong, reward-driven eating impulses, leading to faster eating rates, more frequent food-seeking behaviors, and difficulty terminating consumption even after satiety is reached [56]. At the neural level, this may be attributed to diminished top-down regulation of reward-related regions in the prefrontal cortex caused by weakened inhibitory control, resulting in heightened and poorly regulated responses to food cues [57]. In parallel, individuals with RRBs commonly exhibit challenges in updating and maintaining working memory, which compromises their ability to monitor internal hunger and satiety cues in real time. This weakened satiety monitoring may further promote food approach behaviors by disrupting the feedback mechanisms that normally signal meal termination [58]. Over time, chronic working memory deficits may also impair adherence to long-term dietary self-regulation and health-related goals [59]. Collectively, deficits in both inhibitory control and working memory form two converging pathways that jointly amplify the frequency and intensity of food approach behaviors in children with ASD.

RRBs are closely linked to cognitive rigidity [60], which may manifest in eating behaviors as reduced flexibility in food selection. Children with ASD often struggle to try new foods or adopt different eating styles, showing strong resistance to changes in their established dietary routines. This cognitive rigidity parallels the impaired set-shifting seen in anorexia nervosa [61], although the developmental and clinical contexts of these conditions are distinct. Enhancing executive function—particularly cognitive flexibility—has been shown to reduce eating disorder symptoms by approximately 18% [62], underscoring the potential role of executive dysfunction related to RRBs in shaping eating behaviors. Furthermore, deficits in another domain of executive function, attentional control, may aggravate these difficulties. Specifically, RRBs can reduce the ability of neurodevelopmental children to inhibit avoidance responses to non-preferred foods, thereby increasing their vulnerability to disordered eating behaviors [63].

In summary, the “RRBs–Executive Function–Eating Behavior” pathway proposes that RRBs may serve as a crucial underlying factor contributing to diverse eating behavior problems. Individuals with prominent RRBs often exhibit heightened sensory-seeking tendencies, reward dependence, and cognitive rigidity, all of which play pivotal roles in shaping their eating patterns. Executive function, acting as a potential mediator, may further amplify or modulate these behavioral tendencies through multiple cognitive mechanisms, ultimately giving rise to either food approach or food avoidance phenotypes. Notably, these two behavioral tendencies can coexist and dynamically interact within the same individual, This integrative framework helps explain the complex and seemingly paradoxical phenomenon observed in many children with ASD, who may simultaneously “eat only a limited range of foods” yet “overeat those preferred foods”.

### 4.2. The Influence of Social Deficits on Eating Behavior in Children with ASD and the Mediating Role of Executive Function

This study found no direct association between social deficits and eating behaviors in children with ASD. This result may suggest that the motivational architecture driving eating behaviors differs fundamentally between individuals with ASD and those with TD. In TD populations, psychosocial factors such as social anxiety, peer pressure, interpersonal dynamics, and body image concerns often serve as primary motivators for maladaptive eating behaviors, including restrictive eating and emotional overeating [64,65,66]. In contrast, individuals with ASD commonly display impairments in social cognition and diminished sensitivity to social evaluation [67], which may partially buffer them from this dominant socio-emotional pathway. Consequently, their relative insensitivity to social norms may lessen the impact of peer comparison and sociocultural appearance pressures on eating behaviors, thereby providing a reasonable explanation for the absence of a significant direct effect observed in this study.

Furthermore, among TD children, individuals are often motivated to modify their behaviors—including eating habits—to integrate into social groups and maintain a sense of social belonging. In contrast, individuals with ASD tend to show diminished sensitivity to social rewards such as smiles and praise, while maintaining or even exhibiting heightened responsiveness to non-social, sensory-based rewards [68]. As a result, eating behaviors in ASD are primarily influenced by immediate sensory preferences, the intrinsic rewarding properties of food, and a pronounced need for sameness that helps reduce uncertainty. These motivational mechanisms may function largely independently of social deficits, highlighting a distinct motivational architecture underlying eating behaviors in ASD.

However, the absence of a direct effect does not suggest that the social dimension is irrelevant to eating behaviors in individuals with ASD; rather, it indicates that the mechanisms of influence may differ from those observed in TD populations. We propose that, in ASD, social deficits likely affect eating behavior through cognitive processes rather than socio-emotional ones. In TD individuals, anxiety triggered by concerns about negative social evaluation can directly lead to behavioral regulation of eating, such as dieting or binge eating. In contrast, individuals with ASD tend to display reduced sensitivity to such social threats, which may explain the disruption of this socio-emotional pathway. Consequently, the present findings underscore the need to adopt a cognitive framework to better understand how social functioning relates to eating behaviors in ASD.

Although the direct pathway between social deficits and eating behaviors was not supported in this study, the mediation model revealed a significant indirect effect, suggesting that social deficits may affect eating behaviors via the mediating influence of executive function. Research has demonstrated a close and potentially bidirectional relationship between social competence and executive function, particularly in metacognitive domains such as working memory, cognitive flexibility, and planning ability [69]. Individuals with ASD often experience heightened stress and physiological arousal (e.g., elevated cortisol levels) when navigating social situations [70]. One explanation is that processing complex social information requires substantially greater cognitive effort. This sustained cognitive demand may continuously tax and deplete core executive resources—particularly working memory and cognitive flexibility [71]. Such chronic depletion, in turn, may impair their ability to regulate behaviors that rely on cognitive control, including eating. Consequently, social deficits may indirectly influence eating behaviors by “overdrawing” the already vulnerable cognitive control systems of children with ASD. Nonetheless, the specific cognitive depletion mechanisms involved remain to be empirically validated in future research.

In summary, social deficits may exert comprehensive indirect effects on eating behaviors through the mediating role of executive function. The establishment of this “Social Deficits–Executive Function–Eating Behavior” pathway resonates with the previously discussed “common pathway” model, which posits that RRBs influence eating behaviors via executive function. Within individuals with ASD, regardless of whether the core symptoms are social or non-social in nature, executive function consistently emerges as a pivotal intermediary mechanism shaping complex behaviors such as eating. By integrating the influence pathways of both deficits domains into a unified cognitive framework, this study provides a robust theoretical foundation for developing executive function–based interventions to address the heterogeneous behavioral profiles observed in children with ASD.

### 4.3. Absence of BMI Differences and Possible Explanations

An unexpected finding of this study was that children with ASD did not exhibit significant differences in eating behaviors across BMI categories. This contrasts with previous research linking BMI to behaviors such as picky eating in ASD populations [72,73], underscoring the complexity and heterogeneity of diet–weight relationships in this group. Several factors may help explain this pattern. Food-approach and food-avoidance tendencies often coexist in individuals with ASD [74], and their opposing metabolic effects may counterbalance one another—for instance, attraction to high-calorie foods may coincide with avoidance of many other foods. Moreover, BMI is shaped by numerous influences beyond eating behavior, including basal metabolic rate, physical activity, medication use, and the family feeding environment [75]. The cross-sectional design of this study further limits the ability to capture the dynamic interactions among these factors. Additionally, the substantial heterogeneity within the ASD population may dilute potential associations.

Taken together, these considerations suggest that the relationship between eating behaviors and BMI in children with ASD may not follow a simple or direct pathway. The path analysis in this study indicates that executive function may operate as a key cognitive mechanism linking ASD core deficits to atypical eating behaviors. Although this pathway did not extend to BMI differences in the current cross-sectional design, it offers a useful framework for understanding the complex and heterogeneous eating behavior patterns observed in ASD. Future research should employ longitudinal designs and multidimensional assessments to clarify how this pathway interacts with individual and environmental factors to influence long-term health outcomes.

## 5. Limitations

This study has several limitations. First, all core variables—including ASD core symptoms, executive function, and eating behaviors—were assessed solely through parent reports. Although this approach is common in child research, reliance on a single informant may introduce common method bias and inflate associations among variables. Second, the study did not systematically screen for or control co-occurring intellectual disability. Because intellectual disability can affect both executive function development and eating behavior patterns, its omission may have introduced confounding effects. Third, the sample consisted of children aged 3 to 12 years, and developmental variability within this broad age range may have influenced the results. Moreover, all participants were Chinese and of Han ethnicity. While this homogeneity reduces within-sample variability, it limits the generalizability of the findings across cultural and ethnic contexts. Finally, the cross-sectional design prevents any causal interpretation. Future research should therefore increase sample size and diversity, employ measurement tools tailored to specific developmental stages or behavioral domains, and incorporate contextual variables (e.g., eating environment) when examining eating behaviors. Furthermore, more systematic, age-stratified longitudinal studies are required to elucidate the causal mechanisms underlying these associations and to clarify the model’s potential developmental trajectories.

## 6. Conclusions

This study investigated the factors contributing to eating behavior problems in children with ASD and provided valuable insights into the associations between core autistic symptoms, eating behaviors, and their underlying mechanisms. The findings revealed that RRBs, one of the core characteristics of ASD, were associated with eating behavior problems, whereas social deficits showed no significant direct relationship with eating behaviors. Moreover, executive function emerged as a potential mediating mechanism linking core symptoms to eating behavior outcomes. While additional research is required to clarify the neural underpinnings of these associations and to confirm the observed relationships through longitudinal designs, the present study establishes a theoretical basis for improving eating behaviors in children with ASD and identifies executive function as a promising target for individualized behavioral interventions.

## Figures and Tables

**Figure 1 nutrients-17-03854-f001:**
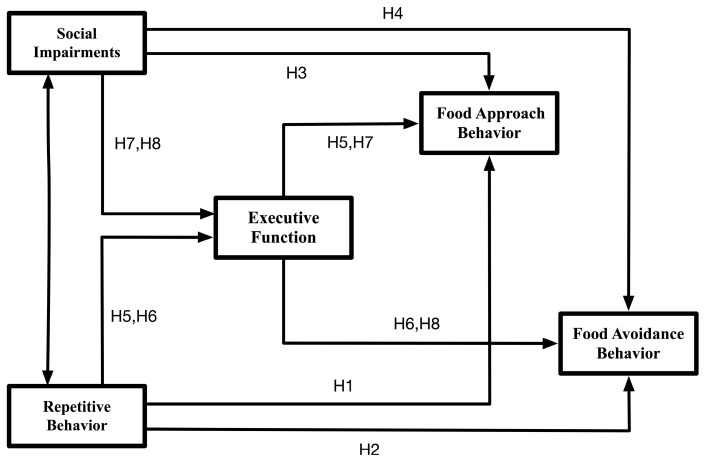
Theoretical model of the research variables.

**Figure 2 nutrients-17-03854-f002:**
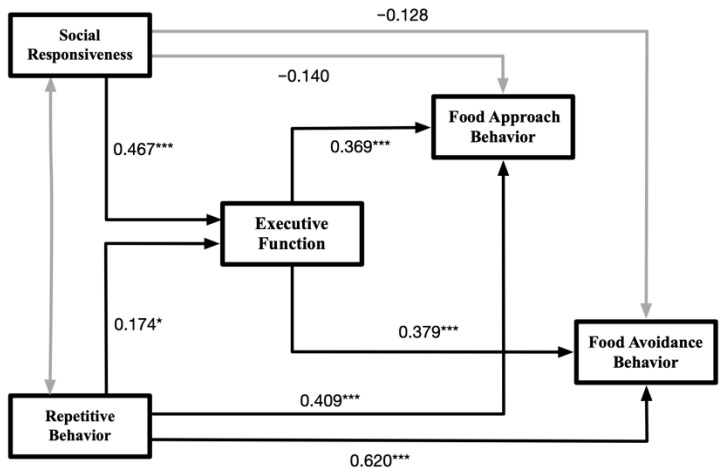
Schematic diagram of the results of the path modeling test. Notes: The figure only displays the relationships between the latent variables. The values on the paths represent standardized coefficients, with black paths indicating significance and gray paths indicating non-significance. Significance levels: * *p* < 0.05, *** *p* < 0.001.

**Table 1 nutrients-17-03854-t001:** Participants’ Characteristics and Between-Group Comparisons.

Characteristic (Mean ± SD)	Category	Frequency (n)	Percent (%)	SRS ^a^ (Mean ± SD)	RRBs ^b^ (Mean ± SD)	EF ^c^ (Mean ± SD)	FAP ^d^ (Mean ± SD)	FAV ^e^ (Mean ± SD)
Gender	Male	151	82.07	2.36 ± 0.36	1.63 ± 0.34	3.44 ± 0.59	2.43 ± 0.58	2.58 ± 0.45
	Female	33	17.93	2.45 ± 0.36	1.56 ± 0.29	3.62 ± 0.54	2.56 ± 0.52	2.51 ± 0.39
Age (years) (6.42 ± 2.35)	3–5	85	46.20	2.44 ± 0.35	1.61 ± 0.35	3.51 ± 0.51	2.43 ± 0.56	2.57 ± 0.42
	6–8	56	30.43	2.32 ± 0.37	1.58 ± 0.28	3.52 ± 0.59	2.49 ± 0.62	2.56 ± 0.45
	9–12	43	23.37	2.33 ± 0.35	1.70 ± 0.33	3.35 ± 0.70	2.46 ± 0.54	2.58 ± 0.46
BMI ^f^ (kg/m^2^) (17.13 ± 3.35)	<15	44	23.91	2.44 ± 0.39	1.65 ± 0.36	3.48 ± 0.56	2.44 ± 0.61	2.54 ± 0.46
	15–17 ^h^	72	39.13	2.40 ± 0.33	1.64 ± 0.35	3.46 ± 0.58	2.50 ± 0.50	2.58 ± 0.40
	17–20	38	20.65	2.33 ± 0.37	1.55 ± 0.25	3.53 ± 0.58	2.47 ± 0.60	2.53 ± 0.45
	>20	30	16.30	2.29 ± 0.36	1.62 ± 0.32	3.44 ± 0.65	2.34 ± 0.66	2.63 ± 0.49
CARS ^g^ (38.51 ± 6.06)	30–36.5	53	28.80	2.12 ± 0.37 ***	1.43 ± 0.26 ***	2.80 ± 0.30 ***	2.25 ± 0.66 **	2.30 ± 0.39 ***
	≥37	131	71.2	2.48 ± 0.30 ***	1.70 ± 0.32 ***	3.75 ± 0.43 ***	2.54 ± 0.51 **	2.68 ± 0.41 ***

Notes: ^a^ SRS: Social Responsiveness, assessed by SRS-2; ^b^ RRBs: Repetitive behaviors, assessed by RBS-R; ^c^ EF: Executive function, assessed by CHEXI; ^d^ FAP: Food approach, assessed by CEBQ; ^e^ FAV: Food avoidant, assessed by CEBQ. ^f^ BMI: body mass index. ^g^ CARS: Childhood Autism Rating, assessed by CARS; ^h^ Excluding 17. Significance was determined by one-way ANOVA for different groups within the same characteristic. Significance levels are marked as * for ** *p* < 0.01, *** *p* < 0.001.

**Table 2 nutrients-17-03854-t002:** Fit indices of measurement and structural model.

Fit Indices	χ^2^/*df*	SRMR	RMSEA	GFI	AGFI	IFI	CFI	TLI
Allowable Range	≦3.000	<0.080	<0.080	>0.800	>0.800	>0.800	>0.800	>0.800
Goodness of Fit	1.828	0.076	0.067	0.840	0.801	0.890	0.888	0.872

**Table 3 nutrients-17-03854-t003:** Linear regression analysis of the association between RRBs subtypes and eating behaviors.

Eating Behaviors	RRBs	B	SE	β	t	*p*	VIF	95% CI	
								Lower	Upper
Food avoidance behaviors	Stereotyped Behavior	0.145	0.047	0.217	3.106	0.002 **	1.23	0.053	0.237
	Self-Injurious Behavior	0.155	0.084	0.155	1.854	0.065	1.76	−0.010	0.320
	Compulsive Behavior	0.170	0.073	0.173	2.327	0.021 *	1.38	0.026	0.314
	Ritualistic Behavior	0.061	0.087	0.059	0.702	0.484	1.75	−0.110	0.232
	Sameness Behavior	0.075	0.106	0.064	0.714	0.476	2.04	−0.133	0.284
	Restricted Interests	0.104	0.062	0.132	1.677	0.095	1.55	−0.018	0.227
Food approach behaviors	Stereotyped Behavior	0.266	0.065	0.305	4.073	<0.001 ***	1.23	0.137	0.395
	Self-Injurious Behavior	0.037	0.117	0.028	0.315	0.753	1.76	−0.194	0.268
	Compulsive Behavior	0.180	0.102	0.140	1.763	0.080	1.38	−0.021	0.382
	Ritualistic Behavior	−0.072	0.122	−0.053	−0.596	0.552	1.75	−0.312	0.167
	Sameness Behavior	0.189	0.148	0.124	1.279	0.203	2.04	−0.103	0.482
	Restricted Interests	0.082	0.087	0.079	0.937	0.350	1.53	−0.090	0.253

Notes: B: unstandardized regression coefficient; SE: standard error; β: standardized regression coefficient; t: t-value; *p*: *p*-value; VIF: Variance Inflation Factor; 95% CI: 95% confidence interval. Significance levels: * *p* < 0.05, ** 0.001 < *p*< 0.01, *** *p* < 0.001.

**Table 4 nutrients-17-03854-t004:** Intermediation effectiveness test analysis form.

Effects	Path Relationship	Point Estimate	Product of Coefficient		Bootstrapping	
			S.E.	Z	Lower	Upper
Direct Effects	RRBs → FAP	0.855	0.270	3.167	0.416	1.469
Indirect Effects	RRBs → EF → FAP	0.134	0.090	1.489	0.004	0.375
Total Effects	RRBs → FAP	0.989	0.285	3.470	0.546	1.690
Direct Effects	RRBs → FAV	0.750	0.194	3.866	0.435	1.187
Indirect Effects	RRBs → EF → FAV	0.080	0.055	1.455	0.001	0.233
Total Effects	RRBs → FAV	0.830	0.205	4.049	0.494	1.302
Direct Effects	SRS → FAP	−0.316	0.243	−1.300	−0.818	0.152
Indirect Effects	SRS → EF → FAP	0.389	0.159	2.447	0.138	0.767
Total Effects	SRS → FAP	0.073	0.224	0.326	−0.375	0.510
Direct Effects	SRS → FAV	−0.166	0.143	−1.161	−0.473	0.090
Indirect Effects	SRS → EF → FAV	0.231	0.103	2.243	0.071	0.470
Total Effects	SRS → FAV	0.064	0.131	0.489	−0.209	0.317

Notes: SRS: Social Responsiveness; RRBs: Repetitive behaviors; EF: Executive function; FAP: Food approach; FAV: Food avoidant.

## Data Availability

The datasets generated during and/or analyzed during the current study are not publicly available due to ethical reasons, are available from the corresponding author on reasonable request.

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
