# Peer review of "Core Deficits and Eating Behaviors in Children with Autism: The Role of Executive Function"

_nutrients, 2025, doi:10.3390/nu17243854_

Round 1

Reviewer 1 Report

Comments and Suggestions for Authors

The article entitled “nutrients-3979654_ Core Deficits and Eating Behaviors in Children with Autism: The Role of Executive Function” is submitted to the “Nutrition and Public Health ” section of the journal “Nutrients”.

This cross-sectional study of 184 children aged 3–12 years examines how core features of autism spectrum disorder (ASD)—restrictive and repetitive behaviors (RRBs) and social impairments—are related to eating behaviors, and whether executive function mediates these associations. The path analyzes indicate that RRBs are significantly associated with both food approach and food avoidance behaviors. Executive function partially mediates these relationships. Social impairments show weaker direct effects but are indirectly associated with eating behaviors through executive dysfunction. Overall, the findings suggest that core ASD characteristics contribute to atypical eating patterns, with executive function emerging as an important pathway and a potential target for intervention.

Comments

  • Title: The title is informative and accurately reflects the content of the study.

  • Abstract: The abstract effectively presents the key aspects of the work.

  • Introduction: The introduction outlines the complexity of the topic and presents a hypothesis that is appropriately subdivided given its multifaceted nature. The accompanying figure helps to clarify the conceptual model. However, it would be advisable for the introduction to end with a clear and precise statement of the study's objective.

  • Methods:

– A cross-sectional design is used to examine the theoretical model, which appears to serve as a preliminary step within a larger clinical trial framework.

– The authors should explain the rationale for including such a wide age range (3–12 years), particularly because eating behaviors, parental supervision, and motivational factors differ considerably between preschool-aged children and preadolescents.

  • Results:

– In Table 1, the meaning of all abbreviations should be defined in footnotes, and the units of measurement for all variables should be provided.

– The results themselves are highly interesting and relevant.

  • Discussion: The discussion is well structured and thoughtfully interprets the findings. However, it should explicitly address the limitations inherent in a cross-sectional design, particularly the inability to establish causal relationships.

Reviewer 2 Report

Comments and Suggestions for Authors

Thank you for the opportunity to review this manuscript, which addresses an important and relatively under-explored topic: the relationship between core ASD symptoms, executive functions, and eating behaviors in children. The manuscript is generally well structured and uses validated tools with good psychometric support. However, several aspects across sections require refinement.

In the introduction, the narrative is occasionally repetitive. For example, the authors cite the same scoping review more than once across early paragraphs (page 2, lines 52 and 58).

The section would benefit from a more concise synthesis explaining why previous studies merging core ASD symptoms into a single construct might introduce bias.

The introduction could also be improved by more clearly identifying the study’s novelty: the authors state that few studies address mediation by executive function, but they do not sufficiently differentiate their work from prior cognitive-behavioral models.

The methods section is detailed, but some clarifications are required. First, the sample includes “children with ASD and co-occurring intellectual disability” (page 4, but the manuscript does not specify how intellectual disability was assessed or how its severity might influence executive functioning and eating behavior. Since ID is a major confounder for both variables, this omission should be addressed. Second, multiple scales used are very long, yet the authors chose mean subscale scores as manifest variables. While this choice is reasonable with their sample size, the justification is limited and should refer more explicitly to measurement model constraints. Third, the handling of missing data is mentioned but without reporting the proportion of missingness by variable, which is important because multiple imputation should be justified by a missing-at-random mechanism.

Table 1 shows several significant differences by ASD severity, but the text does not discuss why BMI showed no significant associations, which is unexpected given the literature on food selectivity and overweight in ASD. The authors should also clarify whether CARS severity groups were pre-specified or data-driven, as the cut-off at 40 is justified only briefly. In the path analysis, effect sizes should be interpreted more cautiously; for instance, the standardized coefficient linking RRBs to food avoidance (γ = 0.620) appears unusually high for behavioral data and deserves a sensitivity analysis or at least discussion.

The regression analyses on RRB subtypes are a valuable addition, but the authors should comment on potential multicollinearity among RBS-R subscales, as these constructs are typically intercorrelated. Without collinearity diagnostics, claims of independent prediction may not be robust.

The discussion is sometimes speculative, especially when integrating neurobiological mechanisms not directly assessed in the study. For instance, references to striatal hyperactivation or prefrontal top-down regulation should be framed as hypothetical mechanisms rather than inferred explanations from the present data. The narrative comparing ASD eating behaviors to anorexia nervosa requires caution, as the developmental and clinical pathways differ significantly. The conceptual model linking social deficits to executive depletion is innovative but would benefit from stronger empirical support or acknowledgement of its theoretical nature.

The limitations section needs refinement. The authors should explicitly acknowledge that all measures are caregiver-reported, which significantly increases risk of shared method variance. Additionally, the presence of co-occurring intellectual disability should be discussed as a limitation, given its impact on both executive functioning and eating behaviors.

Round 2

Reviewer 2 Report

Comments and Suggestions for Authors

Thank you fro your valuable work